# Remote Technologies as Common Practice in Industrial Maintenance: What Do Experts Say?

Laura Seiffert [1,*], Jana Sczodrok [1], Javad Ghofrani [2] and Katrin Wieczorek [1]

1 Faculty of Economics, University of Applied Sciences Dresden, 01069 Dresden, Germany
2 Institute of Computer Engineering, University of Lübeck, 23562 Lübeck, Germany
* Correspondence: laura.seiffert@htw-dresden.de

**Abstract:** Remote solutions open up new possibilities for collaboration and communication to solve maintenance tasks efficiently. Currently, there are no concepts to efficiently determine the suitability of such remote technologies for specific production facilities. It is therefore even more important to include current practical experience with remote technologies in industrial maintenance in the development of these concepts. In this way, the specific requirements and needs of the companies can be taken into account. In order to get an updated practical impression of the use of remote maintenance technology in the field of industrial maintenance, an explorative interview study was conducted. The aim of this study is to collect detailed examples from practice in order to be able to set up a model (category system) oriented towards practical focal points. Based on this, targeted representative surveys or practice-oriented experimental designs can be developed better. For this reason, ten interviews were conducted with maintenance experts from the business community. The results show that remote technologies have not yet fully established themselves in business practice. The main problem is the implementation of suitable framework conditions in order to be able to use remote technologies extensively.

**Keywords:** maintenance; industry 4.0; remote technologies; augmented reality

## 1. Introduction

In the course of the ongoing digitisation of production systems, also known as Industry 4.0, the networking and complexity of intelligent production facilities is increasing. As a result, the work tasks and therefore the demands on employees are changing [1]. There is a particular need for planning in the area of maintenance, which generally serves to ensure a fluid production process. At present, unexpected problems still lead to high opportunity costs, which is why foresighted planning, and the creation of failure forecasts are crucial for an efficient maintenance process. The existing lack of skilled workers also poses a challenge for efficient maintenance, as the required skilled employees are often not flexibly available in the event of malfunctions and defects in production facilities [2]. In addition, there are further restrictions due to travel limitations possible, e.g., caused by the COVID-19 pandemic, which can lead to bottlenecks in production and longer machine down-times. Downtime and maintenance are high cost drivers, accounting for about 60 to 70 percent of the production costs incurred [3,4]. To counteract these problems, current research is developing concepts for real-time support in the area of maintenance. The use of remote technologies in particular can create new possibilities for collaboration in maintenance [4]. With the help of such remote connections, maintenance experts can provide support from anywhere and assist machine operators or skilled technicians on site. This can be used to counteract demographic change and the associated shortage of skilled workers in maintenance as well as the longer down-times of production plants resulting from the shortage of skilled workers. This can lead to a decisive contribution to reducing opportunity costs.

The problem is that there is no practical concept for identifying suitable remote strategies for maintenance in intelligent manufacturing systems [5]. Each plant has different

requirements and possibilities for performing maintenance remotely. In order to be able to create a concept and give recommendations for its use within the future, it is necessary to find out which requirements have to be met by the work task, the plant and the performance requirements of the maintenance experts and machine operators. To date, remote maintenance has mainly been considered from a technical point of view and technical feasibility. The human factors and their performance requirements tend to be underrepresented in the studies [6]. When developing a concept for identifying suitable remote maintenance strategies in intelligent manufacturing systems, these human factors are often neglected.

*1.1. Maintenance and the Importance of Communication*

For a clear understanding, it is necessary to clarify what is meant by maintenance. It is used as a generic term for the specific measures required to maintain functionality. These include servicing, inspection, repair and improvement. The aim is to delay the processes of wear, destruction and decay of the units under consideration in order to guarantee the most trouble-free use possible or to avoid use that is fraught with disadvantages [7]. According to DIN 31051 (German Institute for Standardisation) & Strunz [7] maintenance can be described as an overview term, which is thematically supplemented by its four sub-functions.

(i) **Servicing** is the term used to describe measures to delay the degradation of the existing wear stock, i.e., the preservation of the target condition.

(ii) **Inspection** is the examination of conformity of the relevant characteristics of an object, through measurement, observation or functional testing, i.e., the determination and assessment of an actual condition.

(iii) **Repair** is considered to be a physical measure carried out in order to restore the function of a defective object, i.e., the restoration of the target condition.

(iv) **Improvement** includes the administrative and technical assessment and improvement of the target state of an object without changing the original function.

Furthermore, DIN EN 13306 (German Institute for Standardisation) distinguishes between preventive and corrective maintenance. The preventive measures of maintenance and inspection are referred to as "preventive maintenance", while "corrective maintenance" refers to repairs after the occurrence of faults or malfunctions.

When talking about remote maintenance, several levels of communication can be used. A distinction must be made between the following:

- **machine-to-machine communication (M2M)**, which describes the automated exchange of information between technical systems, e.g., machines, with each other or with a central office. Typical applications are remote monitoring and control. M2M links information and communication technologies and forms the IoT [8].
- Furthermore, **human–machine interaction**. Here, the machine is in many cases a computer, digital systems or devices for the IoT that contain information and communication technologies and application or information systems [9]. Given the high complexity of modern production facilities, human–machine communication is facing new challenges. The large amount of data collected increases the demands on visualisation, which must be understandable for both maintenance experts and machine operators on site.
- The third level is **human-to-human communication**. An example of this can be seen in Figure 1; it describes the communication in the event of a malfunction between a maintenance expert and a machine operator via various remote technologies such as tablet, data glasses, smartphone. In terms of remote technologies, this can be divided into different levels.
  - First, information exchange between two or more people via email or messaging service.
  - Second, direct live remote connection with and without image or video transmission via telephone, PC or smartphone.

&ndash; Third, live remote connection using AR technologies via collaboration software. In addition to video, this allows objects to be drawn and superimposed in the viewing area.

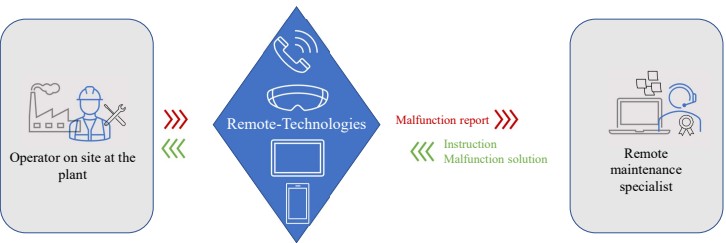

**Figure 1.** Remote Communication.

### 1.2. Aims and Research Questions

Based on a literature research, we have gained knowledge about the theoretical possibilities of remote technologies in industrial maintenance. In this paper, we aim for finding out the practical point of view on using remote technologies in maintenance. In order to achieve the aimed objective, we try to find out exploratory how remote technologies are used in applied practice and which challenges and potentials companies have experienced with regard to these technologies. In order to get an updated practical impression of the use of remote maintenance technology in the field of industrial maintenance, an exploratory interview study was conducted. The aim of this study is to collect detailed examples from practice in order to be able to set up a model (category system) oriented towards practical focal points.

For this purpose, ten manufacturing companies and plant manufacturers were contacted. On the basis of a semi-structured interview guide, they provided information about: 1. the remote technologies used, 2. the framework conditions necessary for the use of remote technologies (e.g., ergonomic requirements), 3. the work tasks in maintenance (scope and complexity) and 4. important human factors such as the performance requirements for the worker or the necessary performance prerequisites to be able to meet these requirements. For example, a machine operator must be qualified enough to be able to exchange technical information with the maintenance expert. Information on the characteristics of these four points enables the development of a suitable remote maintenance strategy. The examples collected can now be used to build up a system of categories. With this, important information can be collected to derive a suitable remote strategy. This reference model represents the result of this study. This should enable an assessment of the use of remote technologies for specific tasks and company requirements. The interviews are not designed to gain insight into the economic efficiency of the use of technology.

We formulated our research questions as following:

- **RQ1: What are examples of successful remote strategies in industrial maintenance for intelligent manufacturing systems (Industry 4.0)?**
- **RQ2: What are the characteristics of the remote strategies?**
- **RQ3: Which categories can be used to describe remote strategies carefully in order to evaluate their success?**

In order to answer these questions, we conducted an exploratory study using semi-structured interviews with experts following a guideline Appendix A.1 as proposed by Döring & Bortz [10].

### 1.3. Structure of the Article

This article is structured as follows. In Section 2, (Related Work) we review the relevant literature to find out to know what knowledge and experience already exists on the use of Industry 4.0 technologies such as VR/AR technology or the digital twin for maintenance in intelligent manufacturing systems. Section 3 (Materials and Methods) presents the details

of the methodology. An exploratory approach is described with which an insight into current usage scenarios and challenges in the application of Industry 4.0 technologies in the practice of maintenance is to be gained. This preliminary study is intended to serve as a basis for more precisely incorporating the requirements from practice in the subsequent main experimental study. Section 4 presents the results. They include responses from guided interviews, which are summarised using categories to answer the research question described in Section 1. The discussion in Section 5 critically places the results in the research context. Finally, Section 6 provides a summary of the article and the outlook for research priorities that emerge from the study.

## 2. Related Work

The intersection between digitisation and maintenance, especially predictive maintenance, in production systems is well investigated recently [11]. Emergence of disruptive technologies for digitisation such as robotic, cloud computing, big data analytic, cyber physical systems, digital twins and machine learning makes it possible to monitor and collect data from production systems and analyse and utilise them for more precisely planning of maintenance. Plenty amount of technical reports from successful application of cyber physical systems for monitoring of production systems in different domains are available. An example of digitisation of agriculture is reported by Zheng et al. [12]. They apply Unmanned Aerial Vehicle, big data analytics and neural networks to monitor oil palms. More related studies are reported by Leukel et al. [13]. Their systematic literature review proposes an overview on application of machine learning for failure prediction in industrial maintenance. However, most of studies and technical reports discuss the benefits of application of digitisation in different domains using individual case studies (e.g., using remote monitoring for integration of digitisation to total production maintenance concept [14]). Despite the existing research on predictive maintenance, there is no systematic examination that addresses the challenges of the topic of maintenance in digital manufacturing systems. It is only described that maintenance should be digitised and implemented with the support of the new Industry 4.0 technologies, but not exactly what this implementation should look like. Another issues which is not covered in the literature is the increasing complexity of production facilities which leads to increased demands on the qualifications of employees. Technical assistance systems of Industry 4.0 for maintenance can be used in particular to support employees in machine monitoring and troubleshooting, so that they can better meet the increasing requirements through continuous further qualification [15]. To date, augmented reality (AR) has been described as a promising technology for assisting employees in industrial settings. However, many of these systems are not developed beyond prototype status [16]. Masoni et al. [17] see one possibility for solving this problem in the connection of experts who convey instructions to unqualified machine operators with the help of AR. The latest solutions from the field of AR are considered to be particularly promising for use in maintenance in order to provide targeted support, e.g., where to look for sources of malfunction or what correlations exist [4,6,18,19]. There are also barriers to the widespread adoption of AR in various and complex industrial processes, e.g., the lack of adaptable and scalable AR work instructions [20] and low number of applications for user-oriented collaboration [21]. Furthermore, combination of Digital Twin and other technologies, e.g., AR are discussed in [22]. Our previous work [23] elaborates the need for remote technologies in industrial maintenance, especially AR, and summarises the challenges on the human, machine and organisational levels that need to be overcome for practical application in regard to literature. In this paper, we primarily address the often neglected human factor, as people are still the deciding factor for the success of maintenance processes despite all automation and digitisation. In addition, we also look at technical and organisational aspects of maintenance. We address the lack of overview on technical challenges and technological aspects for performing maintenance in digitised production systems. As it is still not known which Industry 4.0 technologies are applicable for which application scenario in maintenance and how an effective implementation of the related

technology can proceed. The main contributions of our paper are therefore: (1) to propose a theoretical framework to collect and analyse the state of practice among experts in the field of maintenance, and (2) to show how the theoretical framework can be applied to study the implementation of remote technologies in digitised production systems. We believe that the results of our study will open up further research opportunities to bridge the gap between the technical and theoretical aspects of smart production system maintenance among researchers and practitioners.

## 3. Materials and Methods

As already indicated, an exploratory approach with an interview study was chosen to answer the research questions. We interviewed ten maintenance experts from the field who were able to provide information on both operational processes and strategic considerations. The interviews were conducted as semi-structured interviews. This approach enables a very detailed recording of companies' experiences with the use of remote maintenance technologies. Furthermore, the methodology of content-analytical summarising was chosen, which pursues the goal of reducing the material in a way that the essential content is retained and, through abstraction, creating a manageable corpus that is still a reflection of the basic material [24]. From this detailed information, it is possible to derive which information is necessary to be able to identify an individually suitable remote maintenance strategy for companies. Furthermore, in order to be able to collect this information in a structured way in the future, a category system will be developed from the interview data, which will also be referred to as a reference model in the following.

### 3.1. Interview Guideline

For this purpose, we developed an interview guideline Appendix A.1. This was evaluated by experts from research and practice, tested in mock interviews and adjusted.The aim of the series of interviews was to collect data on the current state of practice as well as on challenges, potentials and prerequisites for the efficient use of remote maintenance solutions. To structure the interview, we first selected introductory questions, followed by key questions and forward-looking questions. Closed questions were used for factual information and open questions for content-related information. The questions were asked as main questions and, if necessary, supplemented by specific follow-up questions. The introductory questions refer to general information about the company, a general technical-organisational transformation and the general influence of digitisation on work areas. The key questions relate specifically to experiences with remote technologies. For this purpose, we specifically asked questions about the process of the individual subareas of maintenance (servicing, inspection, repair, improvement) in order to determine the activities and circumstances of the respective industry as well as differences in the work steps. In the area of future questions, we wanted to find out which potentials the companies see for their company in the future through remote technologies. Further details are provided in the Appendix A.

### 3.2. Data Collection

For the data collection, 26 network contacts from the manufacturing industry were acquired. In the period from January 2020 to March 2020, 11 interviews were conducted, 10 of which were included for the interview analysis. This small number of research units is sufficient, as a qualitative theory-building research approach is followed here to investigate an open research question [10]. The research units (the 10 cases/interviews) are described in great detail. This allows a theory to be built using a catalogue of criteria to evaluate successful remote strategies.

The selected interview partners are experts in maintenance and have extensive knowledge of the maintenance practice of their respective companies. They come from two different industries (semiconductor industry and mechanical engineering) and have different roles in the maintenance process (Managing Director, Facility and Plant Management,

Head of Maintenance and Equipment Engineering, Senior Specialist Factory Integration, Development Engineer, Senior Environmental Engineer, Head of Service Department, Head of Technical and Support Department, Team Manager Incident Management). They represent different technological competences, e.g., condition monitoring, industrial measurement technology, maintenance planning. The diversity of the interviewees in terms of industry type, role in the company and competencies allows a greater differentiation of the perspective on industrial maintenance and thus results in a more detailed insight into the characteristics of different remote strategies (see Research Question 2).

Participation in the interviews was conducted by video call via Microsoft Teams and recorded as an audio file. Two interviews were conducted in person at the companies' locations and could not be recorded. Here, the memory transcripts serve as the data basis for the analysis. The duration of the interviews was between 45 and 90 min.

*3.3. Data Analysis—Qualitative Content Analysis*

For the analysis of the interviews, the method of qualitative content analysis according to Mayring [24] was chosen. With this methodological approach, not only manifest but also latent contents can be scientifically correctly worked out and structured from verbal material [25]. This enables a more detailed analysis of the interview material and, as a result, a more differentiated system of categories for the reference model to be developed. In preparation for the analysis, the interviews were transcribed in a simplified form [26]. The transcripts are written in German and are therefore not attached to the article.

The transcribed material was then categorised using a deductive-inductive approach. In a first step, the following basic categories were derived from the literature:

- Employee Acceptance [27]
- Remote Technologies [28]
- Organisation and framework conditions [27,29]
- Sustainability [30]
- Costs/Benefits [30]

These deductively derived categories were then structured according to the human-technology-organisation approach based on the action regulation theory [31] and the concept of socio-technical systems [32]. This still rough category system formed the basis for structuring the data. During the structuring of the data, the categories were further differentiated (inductive part of the analysis). By using assignment rules, which determine how the individual statements are to be abstracted, the summarised interview statements are assigned to the deductively derived categories. If a categorisation is not possible in terms of content, categories (main or subcategories) are added. These assignments are made in an iterative process in which the category system is repeatedly revised and checked [24].

## 4. Results

In the following, the results of the analysis are presented in the form of the category system developed, consisting of main categories and subcategories. In addition, examples from the data material are assigned to each category in the form of key statements to enable a better understanding of the meaning of the categories. Following these descriptions, the research questions are addressed.

*4.1. Category System*

Based on the methodology explained in the previous section, ten main categories were defined. Two of them have subcategories. Table 1 gives an overview of the category system. In addition, information on the interview partners regarding industry, role and technical competence can be found in the Appendix A.2. In the following, all categories are defined and the core statements identified from the interviews in the qualitative analysis are presented in summarised form.

**Table 1.** Category System.

| Main Category | Subcategory |
|---|---|
| **C1** Physical Safety | |
| **C2** Performance Requirements | **C2a** Technical and Methodological Competence |
| | **C2b** Digital Competence |
| | **C2c** Demography and Technology |
| | **C2d** Human Resource Development |
| | **C2e** Communication Skills |
| **C3** Employee Acceptance (trust/scepticism) | |
| **C4** IT Infrastructure | |
| **C5** Remote Technologies | |
| **C6** IT Security | |
| **C7** Organisation and framework conditions | **C7a** Customer Dependency |
| | **C7b** Sustainability |
| **C8** Costs/Benefits | |
| **C9** Potential use cases | |
| **C10** Remaining Category | |

4.1.1. Category 1—Physical Safety

The inductively developed category Physical Safety contains statements by the interviewees on potential hazards that may arise from the use of remote technologies in industrial maintenance.

From the core statements of the category, the following can be summarised: While the machine operator follows the instructions of the remote expert, their safety at the workplace cannot be guaranteed due to the remote expert's limited perception of the environmental influences (Interview A, H). This also includes the fact that due to a lower level of training of the machine operator, it cannot be assumed that he/she is able to overlook all possible safety risks during the instructed tasks. Not to mention that additional risks can arise due to a lack of technical knowledge about remote technology (Interview E).

4.1.2. Category 2—Performance Requirements

This category describes, with the help of five subcategories, the necessary performance requirements on the aspects of technology and methodology, digital competence, demographics, personnel development as well as communication skills needed by employees in the maintenance process via remote maintenance.

- Subcategory 2.1 *Technical and Methodological Competence* describes all statements on required expertise on maintenance, processes and concrete machines.

From the core statements of the category, the following can be summarised: Machine operators are not sufficiently qualified to carry out service, inspection or repairs on their own due to their existing work tasks. Technical expertise or special training is required to carry out these various maintenance tasks. Machine operators are only able to perform sub-tasks with low complexity, such as localisation of a malfunction in the superficial machine area. In order to be able to transfer tasks from the different areas of maintenance by means of remote technologies, it must be checked individually whether these can be carried out without further special knowledge available on site (Interview H), for example by classifying the tasks according to difficulty levels (Interview G).

- Subcategory 2.2 *Digital Competence* contains statements about required technical knowledge and confidence in the use of remote technologies.

Machine operators need to have the ability to set up the remote-technologies as well as the necessary software and secure connection as a basis for successful remote support. The correct handling and operation of the hardware is a challenge for the efficient use of new technologies (Interview A). In the Future, this will have a higher relevance for new positions in smart manufacturing (Interview F).

- Subcategory 2.3 *Demography and Technology*, states all statements related to the use of technology or technical affinity and age.

Acceptance of new technologies and process changes is more common among the younger generation of employees (Interview B), moreover, younger employees adapt to new technologies more quickly. Their ability to learn how to use technologies is also higher than that of older employees.

- Subcategory 2.4 *Human Resources Development* includes all statements related to needed training and education of employees due to the use of remote technologies.

When introducing new technologies, training is urgently needed so that employees understand the new processes in detail and work safety can be guaranteed. With additional training and education as well as individual case assessment, machine operators could take over simple maintenance tasks (Interview G). The use of data glasses and AR technologies alone requires extensive training and practice (Interview G). It is assumed that due to the increase in automation in manufacturing, there will be fewer machine operators on site in the future. The demands on the remaining workers will therefore be higher in the future and further qualification in the area of maintenance and technology will be necessary (Interview F).

- Subcategory 2.4 *Communication Capability* refers to all statements on the efficient exchange of information via remote technologies and the influence of the technical know-how of sender and receiver.

A prerequisite for the instruction of a person via remote is corresponding know-how on the part of both or all communication partners. A task transfer can only be instructed remotely to another expert, since a standardised interface must be available for successful communication (Interview B). This means that the operator must have special expertise in order to understand processes, machine settings and complex technical interrelationships (Interview B).

### 4.1.3. Category 3—Employee Acceptance

This category includes statements about employee mindset and acceptance of new solutions, procedures and general changes to existing processes. The focus here is also on employee motivation, existing scepticism or lack of trust regarding new technology, people and processes.

From the core statements of the category, the following can be summarised: In order to successfully introduce remote technology, it is urgently necessary to reduce existing fears and doubts of all participants in the maintenance process, especially on the topics of IT security and effectiveness of the new process, which are related to the change process.

### 4.1.4. Category 4—IT Infrastructure

This category includes all statements from the interviews regarding the IT infrastructure required for remote maintenance. This includes aspects such as the availability of WiFi, bandwidth and stability of the network, equipment and selection of suitable technologies.

From the core statements of the category, the following can be summarised: In order to be able to carry out remote maintenance successfully, technical framework conditions such as sufficient WiFi coverage throughout the company, a high data bandwidth and network stability are essential. It is also often not possible to use the technologies properly, as the software used is often developed quickly and can contain many errors at the beginning (Interview F, I). The technologies needed for remote maintenance would also have to be available on both communication sides. For emerging technologies, such as data glasses,

there is not yet a large market and the acquisition costs for these can be very high. This could create a financial barrier for companies that cannot afford the necessary investments and running costs, such as licences for AR software. In addition, the lack of a standard for software and hardware poses a major challenge for the efficient use of remote solutions. By standardising the software and hardware used, processes would be better monitored and adjustments could be made remotely (Interview I).

### 4.1.5. Category 5—Remote Technologies

This category contains all statements of the respondents regarding remote technologies and providers in their own companies.

From the core statements of the category, the following can be summarised: To date, classic remote technologies have been used in companies for remote support and communication. The exchange of information on simple maintenance problems usually takes place via telephone, e-mail or video connection using TeamViewer or Skype. However, there are often limits to these types of customer support. If the complexity of the problem increases, maintenance is carried out on site by a maintenance employee. Remote maintenance, is currently only carried out sporadically for software problems. Technologies such as data glasses to use AR and VR functions have already been acquired by some companies, although these are still being tested (Interview A, F, G). The Microsoft HoloLens 2 as well as the HMT-1 from RealWear are used for testing scenarios, which were chosen due to the criteria VR/AR experience, weight and low acquisition costs (Interview A, E, G). The first findings from the companies' tests are the occurrence of motion sickness during use which can lead to headaches and nausea, among other things (Interview A). Additionally, the user's field of vision is limited and the labour law factors and influences are still unclear, as there are no extensive use cases from research in this area yet (Interview G).

### 4.1.6. Category 6—IT Security

The category IT security is described through all statements on challenges and problems related to data security when using remote solutions.

Summarised from the core statements, the following characteristics emerge for the category: The assessment of security risks is very individual and must be carried out on a single case basis. Customer companies in particular would fear higher risks from remote connections than from on-site maintenance work with cable connections (Interview B). The background to the concerns of some customers could also be the industry in which they work. Especially in the semiconductor industry, important company secrets are contained in the data, such as product definitions, also called "recipes" (Interview G). Lack of standardisation of technologies is another obstacle to guaranteeing IT security during the remote connection (Interview H).

### 4.1.7. Category 7—Organisation and Framework Conditions

The use of remote technologies is not only influenced by human or technical aspects. Organisational factors and circumstances can affect the (efficient) use of remote solutions as well.

- Subcategory 7.1 *Customer Dependency* is described by statements regarding decisions, chosen methods in maintenance, as well as wishes or requirements for maintenance and repair services from the customer side.

The role of the customer and the demand for remote maintenance have an increased influence on whether machine manufacturers offer remote maintenance services. The value of on-site maintenance is rated higher than that of remote maintenance. Accordingly, customers are not willing to pay the same price for both services (Interview B). The need for remote maintenance is rated less in the areas of maintenance and inspection than in the area of repair. In terms of customer dependency on remote maintenance, the cost of remote solutions is an important factor. They depend on the customers' willingness to pay and their general service requirements (Interview G). In addition, the required technologies,

software or licences must also be purchased by the customer, which is why barriers to the implementation of remote solutions quickly arise (Interview H). The aspect of the lack of standardisation of the software used, which has already been mentioned twice in previous categories, is also related to customer dependency. It is often not clear whether the required software can be installed on the customer side. Furthermore, many companies insist on using their own software (Interview H).

- Subcategory 7.2 *Sustainability* includes statements on the perceived awareness of sustainability in the company.

Sustainability is an important issue for the companies surveyed. In particular, preventive maintenance is mentioned here to counteract unplanned machine down-times. However, the principle and the implementation of preventive maintenance is not yet practised in any of the companies.

### 4.1.8. Category 8—Costs/Benefits

In this category all statements that refer to the consideration of costs and benefits regarding the implementation of remote technologies are stated. The focus here is on whether technologies can be used in the company and offer a cost advantage.

As already mentioned in the category of customer dependency, customer companies often do not realise the benefits of maintenance in general. As a prerequisite for the introduction of remote maintenance solutions, awareness must be created. In particular, the aspect that machine downtime is often more cost-intensive than preventive maintenance is often not considered. However there is a great potential for remote maintenance solutions, as in many cases, e.g., for minor malfunctions, travel costs can be avoided. However, the high costs of the new technical solutions would be a barrier for many companies (Interview D).

### 4.1.9. Category 9—Potential Use Cases

This category collects all statements about existing or future use cases for remote maintenance.

Summarised from the core statements, the following characteristics emerge for the category: Currently, the companies surveyed do not have any use cases for the application of remote technologies in industrial maintenance. However, potentials for the use of remote technologies are seen in the areas of improvement and quality control, visualisation, monitoring and fault diagnosis as well as in the improvement of the visualisation of machine states (Interview B, C). Potentials for remote maintenance are also seen above all in remote support of already highly qualified personnel (Interview D). In this case, however, tasks could only be instructed to another expert and not to a machine operator, since a standardised interface must be available for successful communication between the instructor and the instructed. A further potential for communication via AR glasses was seen in the future in order to shorten distances between the clean-room and the non-clean-room, which is particularly relevant for semi-conductor technology. The option of instructing a less qualified machine operator to carry out a complex fault solution is not seen as feasible by any of the companies interviewed.

### 4.1.10. Category 10—Remaining Category

The last category includes statements that are important in terms of content and context, but cannot be assigned to any other category. In this case, this is all background information on the companies that serves as contextual knowledge for the classification of the respective statements.

When it comes to maintenance in industrial production, there is no single way to implement it. Therefore, when implementing maintenance scenarios, it is important to consider who is responsible for the maintenance of the equipment in the company, what possibilities are offered by external maintenance providers, and what prerequisites must be fulfilled by both sides so that efficient and safe remote maintenance can be implemented.

In some cases, machine conditions are viewed online via remote access or video telephony is used for consultation purposes. However, no repair measures are carried out remotely under the guidance of an expert.

*4.2. Answering the Research Questions*

Based on the core statements in the different categories, the research questions will be answered.

**RQ1: What are examples of successful remote strategies in industrial maintenance for intelligent manufacturing systems (Industry 4.0)?**

From the core statements, it can be summarised that remote solutions are not yet primarily used for maintenance in the surveyed companies. Currently, industrial maintenance tasks are mostly performed on-site at the plants and by trained specialist personnel. The instruction of an employee who has not been specially trained by a maintenance expert does not yet take place in practice. Looking at the sub-areas of maintenance, it can be stated that remote technologies are used sporadically in the areas of inspection and servicing, but this does not yet take place in the area of repair. It should be noted here that the distinction between the technical terms was not always clearly delineated by some interview partners. For example, terms such as "maintenance" and "servicing" were occasionally used as synonyms by the interviewees, as the delineation of the different areas of maintenance cannot always be clearly distinguished from each other in practice. Referring to the stages of human-human communication which have already been introduced, the first two stages of human-human communication via remote are often used in the case of problems with machines. Above all, e-mail and telephone are common methods of contacting manufacturing companies in the event of malfunctions. Although video connections (e.g., via TeamViewer) are also used in addition to e-mail and telephone, the problem is rarely solved in this way. These types of remote support serve to identify faults and to make an initial assessment of the complexity of the problem. Scenarios in which faults are solved remotely are a rarity and are limited to software errors or changes that can be fixed remotely on machines. For the use of data glasses in combination with collaboration software and AR technologies, no explicit use cases and practical experience reports can yet be presented. AR solutions are currently being tested by three of the companies surveyed with regard to their suitability for usability and costs.

**RQ2: What are the characteristics of the remote strategies?**

Based on the literature and data material, the following characteristics for remote maintenance strategies can be stated. As illustrated in Figure 2, they are sorted according to the human-technology-organisation approach:

4.2.1. Human

- **Physical safety for the working persons:** In intelligent manufacturing systems, hazards to the physical health of the machine operator can arise both during regular operation and in the event of a malfunction. When implementing a remote maintenance strategy, these hazards must be reassessed. It must be assessed to what extent they change through the implementation of remote processes and whether the existing personal safety measures such as training and personal protective equipment are still appropriate.
- **Qualification of the machine operator:** Intelligent manufacturing systems are usually complex in structure, functionality and operation. The implementation of a remote maintenance strategy can change the requirements for the qualification of both the machine operator and the maintenance staff. It must be examined how the activities of the machine operator and the maintenance expert change, among other things with regard to the work task as well as the use of technical aids, and whether a qualification gap arises as a result.

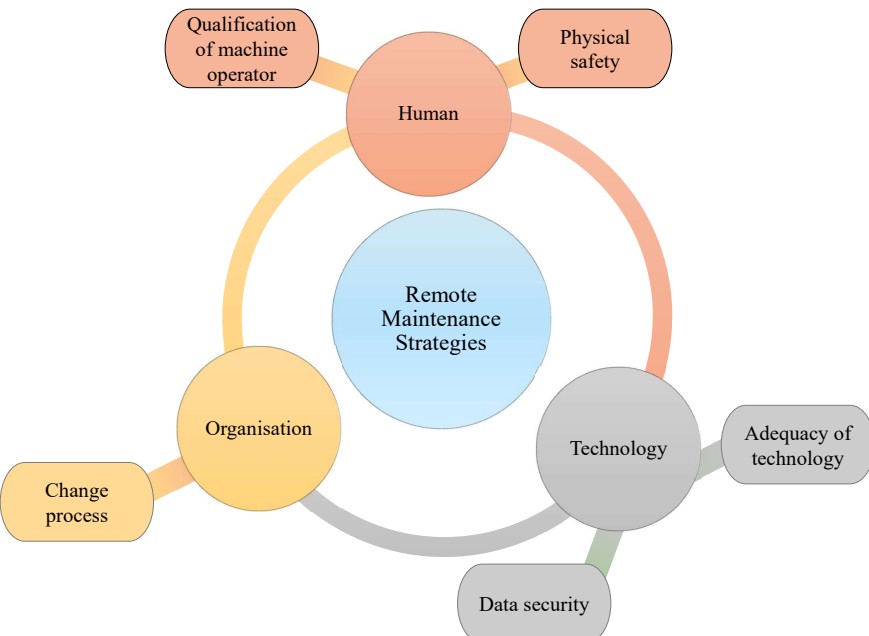

**Figure 2.** Characteristics for remote maintenance strategies.

4.2.2. Technology

- **Data security:** Production systems generate, store and send security-relevant information. This information flow can be influenced by implementing a remote maintenance strategy. It is necessary to check which information reaches external information processing systems (e.g., cloud-based applications) through the remote process via interfaces and whether these meet the company's security requirements.
- **Adequacy of technology:** Intelligent manufacturing systems vary widely in structure and functionality. This results in different requirements for the remote maintenance technology needed. For the development of a suitable remote maintenance strategy, the technical characteristics of the manufacturing system must therefore be identified and used as the basis for deciding on the appropriate remote technology. In addition, the current remote technology is not yet able to transmit all signals of the manufacturing system (noise, odors, vibrations). It is necessary to check whether information about such signals is needed for the maintenance process. Furthermore, it is a challenge to select the appropriate remote technology. Knowledge of the current developments in remote technology variants is usually not available in sufficient detail in the companies to be able to make an informed purchasing decision. Further aspects to be considered from a practical point of view are stability of the connection, bandwidth or loss of information during transmission.

4.2.3. Organisation

- **Change process:** The implementation of a remote maintenance process is a change process. During this process, not only is new technology implemented, but work tasks, workflows, rights/responsibilities, and social interaction are changed, among other things. These changes can lead to acceptance problems for the work people involved. To avoid such problems, a suitable change management concept (participation, transparent communication of plans and measures) should be part of the implementation strategy.

**RQ3: Which categories can be used to describe remote strategies carefully in order to evaluate their success?**

In order to evaluate the success of remote maintenance processes, an overview of the most relevant evaluation categories is needed. Through literature research and interview

studies, these categories were compiled in this article. A detailed description of these categories can be found in Section 4.1 Category System. This overview can help users to systematically build up a remote maintenance strategy and to evaluate it with regard to its probability of success.

## 5. Discussion

Although remote technologies such as data glasses have been on the market for several years, the companies surveyed have only been using them for testing purposes so far. The reason for this is that the companies surveyed have not yet been able to identify any suitable use cases in their current day-to-day business.

The use of remote technologies for troubleshooting is also considered as too unsafe, as the complexity of the malfunctions to be resolved is considered too high. An efficient malfunction solution can currently only be implemented on site. Through their own test scenarios, the companies surveyed were nevertheless able to name the most important conditions, challenges and potentials. These were elaborated in this article as characteristics of remote maintenance strategies. However, the most fundamental problem in identifying a suitable remote maintenance strategy is the lack of a detailed overview of the constantly developing remote technologies and the corresponding suitability with the existing equipment in the company. The results in this article were derived methodologically correct. They are based on a small sample. This is nevertheless sufficient to provide a well founded answer to the research questions, even if the results would be more reliable with a more extensive data base.

## 6. Conclusions

In this paper, an exploratory interview study on the current use of remote technologies in industrial maintenance practice was conducted. The aim was to use a qualitative approach to establish categories that could be used to evaluate a successful remote maintenance strategy. Through the qualitative interviews, current challenges and potentials of remote maintenance concepts in applied practice could be collected. Based on the literature research and the data from the interviews, all research questions could be answered soundly. It should be noted that there are currently no scenarios for the use of remote technologies in industrial maintenance within the companies surveyed. However, there is potential for the use of remote technologies for activities with a low degree of complexity as well as fault diagnosis, where a maintenance technician receives remote support from another expert on site and training new or inexperienced employees. Based on the findings from the literature research and the interview data collected, the characteristics of remote strategies were also established, e.g., the reduction of scepticism among users, the expansion of the IT infrastructure in companies and the training of employees regarding new technologies as well as the required expertise to carry out maintenance tasks. These were systematically classified with the help of the holistic approach of the interaction of human, technology and organisation. With the answer to the third research question, a reference framework for the success assessment of remote maintenance strategies could be established. The frame of reference consists of categories that relate to human performance requirements as well as technical and organisational framework conditions. This allows for a holistic assessment using the developed reference framework. Moreover, it helps users to systematically build up a remote maintenance strategy and to evaluate it with regard to its probability of success.

In summary, this paper is a first step towards exchanging knowledge and experience on digital maintenance between industry and research. The approach we propose and the resulting evaluation framework can be used by researchers, equipment manufacturers and manufacturing companies that perform industrial maintenance in their daily practice. Both can be used to evaluate and standardise remote industrial maintenance strategies. Thus, this paper can serve as a cornerstone for (a) a first information overview for future research topics/studies dealing with industrial remote maintenance, (b) the development

and adaptation of methods and tools, and (c) the cooperation of companies that intend to plan or implement a remote maintenance strategy themselves.

Further studies are necessary to deepen the state of the art in the field of remote maintenance in intelligent manufacturing systems. Additionally, the focus of research and development should be on the development of standards for remote technologies and methods in order to address the existing practical challenges in the implementation of a remote maintenance strategy. As we discussed in our findings, security plays a particularly important role in the use of remote technologies. The standard to be developed must take data security into account. The integration of security can be considered as one of the main research directions in future studies.

**Author Contributions:** Conceptualization, K.W. and L.S.; methodology, L.S.; validation, K.W., L.S.; investigation, L.S.; data curation, J.S.; writing—original draft preparation, L.S. and J.S.; writing—review and editing, J.G. and L.S.; project administration, K.W. and L.S.; funding acquisition, K.W. and J.G. All authors have read and agreed to the published version of the manuscript.

**Funding:** This measure is co-financed with tax funds on the basis of the budget passed by the Saxon State Parliament.

**Data Availability Statement:** The data presented in this study are available on request from the corresponding author. The data is not publicly available for data privacy reasons.

**Acknowledgments:** We would like to take this opportunity to thank the ten companies that made time for the exchange despite the ongoing COVID-19 pandemic and the additional workload that this entailed.

**Conflicts of Interest:** The authors declare no conflict of interest.

## Abbreviations

The following abbreviations are used in this manuscript:

| | |
|---|---|
| DT | Digital Twin |
| AR | Augmented Reality |
| IoT | Internet of Things |
| M2M | Machine to Machine |

## Appendix A

*Appendix A.1. Interview Questions/Guideline*

**Table A1.** The following are the questions from the interview guideline.

| | | Question (Main Question, Follow-Up Question) |
|---|---|---|
| Introductory questions | MQ | Please give me a brief overview of your company. In which industry does your company operate? |
| | FQ | What is the core business of your company? |
| | | What are the main products? |
| | FQ | Is your company embedded in a larger company? |
| | | If yes, in what form? |
| | | Does your company cooperate with other companies? |
| | | If yes, in what form? |
| | FQ | To what extent does your company deal with maintenance? |
| | | (maintenance, repair, troubleshooting, predictive maintenance) |
| | | How is maintenance organized? (Inhouse, contracts with suppliers, external parties) |
| | MQ | OPTIONAL |
| | | Have there been any major technical/organisational changes in your company in recent years? |
| | | If yes, which ones? |

**Table A1.** *Cont.*

| | | Question (Main Question, Follow-Up Question) |
|---|---|---|
| | MQ | To what extent does your company deal with digitalisation in maintenance, e.g., work tasks, processes, evaluations or similar in maintenance? |
| | | FQ — Since when and to what extent? |
| Key questions | MQ | How much experience does your company have with remote technologies? |
| | FQ | (A) **If no/little experience:** |
| | | Why? Lack of resources? Cost/benefit aspects? |
| | | What conditions would have to be created in order to use remote technologies? |
| | | (B) **If moderate/ much experience:** |
| | | What concrete experience have you gained? |
| | | Which jobs and tasks have changed? |
| | | What has improved in everyday working life? |
| | | What difficulties have arisen in everyday working life? |
| | | Where do you see potential? |
| | | Has any attempt been made to identity additional skill requirements that have arisen as a result of new technologies? |
| | | If so, how? |
| Future issues | MQ | In which areas of maintenance can you imagine remote solutions in the future? |
| | FQ | What should solutions look like in your company? |
| | | Do you see any security risks in the implementation? |
| | | If yes, which ones? |

*Appendix A.2. Backgroundinformation about Interviewees*

**Table A2.** The following table lists the general information about interviewees.

| Interview | Industry | Role of Interviewee | Technical Competence |
|---|---|---|---|
| A | Semiconductor Industry | Facility and Plant Management | Plant and Equipment Commissioning |
| B | Industrial Automation/Industrial Information Technology IT | Head of Service Department | Maintenance Planning |
| C | System Integration | Senior Specialist Factory Integration | System integration and monitoring |
| D | Mechanical Engineering | Managing Director | Equipment Commissioning, Servicing |
| E | Semiconductor Industry | Head of Maintenance and Equipment Engineering | Industrial Measurement Technology |
| F | Semiconductor Industry | Maintenance Equipment Engineering | Industrial Measurement Technology |
| G | Mechanical Engineering for Semiconductor Industry | Head of Technical and Support Department | Condition Monitoring |
| H | Mechanical Engineering | Team Manager Incident Management | Equipment Commissioning |
| I | Semiconductor Industry | Manager Maintenance Equipement and Engineering | Industrial Measurement Technology |
| J | Special Mechanical Engineering | Development Engineering | Monitoring |

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
