# Peer review of "Remote Technologies as Common Practice in Industrial Maintenance: What Do Experts Say?"

_telecom, doi:10.3390/telecom3040031_

Round 1

Reviewer 1 Report (Previous Reviewer 4)

Compared to the initial version, the authors tried to respond to the reported issues. 

Security is very essential in remote access. A question on this topic (in the future) would have been useful, although the security part  is addressed. From my point of view, the authors have answered to the reported issues,

Author Response

Dear Reviewer,

Thank you very much for the helpful comments. We are pleased that you agree with the changes that have been made. We highlighted the importance of security as an essential issue in future work.

Kind regards
Katrin Wieczorek

Reviewer 2 Report (New Reviewer)

-> Abstract: What is AR in key words? -> Introduction: please highlight the contributions of this paper -> Related works: I think the related work is not enough to describe the existing researches. Please highlight the differences between your work and previous works -> Results: I recommend that you can add some figures to make your paper more understanding. -> Some references could be considered to cite: Tortorella, G. L., Fogliatto, F. S., Cauchick-Miguel, P. A., Kurnia, S., & Jurburg, D. (2021). Integration of Industry 4.0 technologies into total productive maintenance practices. International Journal of Production Economics, 240, 108224. Zheng, J., Fu, H., Li, W., Wu, W., Yu, L., Yuan, S., ... & Kanniah, K. D. (2021). Growing status observation for oil palm trees using Unmanned Aerial Vehicle (UAV) images. ISPRS Journal of Photogrammetry and Remote Sensing, 173, 95-121. Leukel, J., González, J., & Riekert, M. (2021). Adoption of machine learning technology for failure prediction in industrial maintenance: A systematic review. Journal of Manufacturing Systems, 61, 87-96.

Author Response

Dear Reviewer,
Thank you for your helpful questions and comments. Here are our responses to them: 

  1. -> Abstract: What is AR in key words?
    AR means Augmented Reality. We refer to it strongly, especially in chapter two. For a better understanding, we will write it out as a keyword.
  2. Introduction: please highlight the contributions of this paper
    Our contribution is now described a little more clearly in the introduction and should be easier to understand. We have also highlighted it at the end of the related work chapter.
  3. Related works: I think the related work is not enough to describe the existing researches. Please highlight the differences between your work and previous works 
    We restructured the text of Related Work and added the suggested literature. In addition, we highlighted the missing aspects in the related work as difference with our work and used it to make the link between our contribution and the necessity of further research accomplished by highlighting our contribution.
  4. Results: I recommend that you can add some figures to make your paper more understanding
    For a better understanding of our results, we have included Figure 2 in the results chapter. It contains the characteristics of the remote maintenance strategy that we found.

Kind regards
Katrin Wieczorek

Round 2

Reviewer 2 Report (New Reviewer)

The authors have addressed all my issues, and I think it can be published in this journal.

This manuscript is a resubmission of an earlier submission. The following is a list of the peer review reports and author responses from that submission.

Round 1

Reviewer 1 Report

I see that this article does not contain a sufficient contribution to accept it in this journal. Authors should improve the contribution of this article, show their economic benefit and correct spelling mistakes.

Author Response

Reviewer: I see that this article does not contain a sufficient contribution to accept it in this journal. Authors should improve the contribution of this article, show their economic benefit and correct spelling mistakes.

Authors' response: 
Thank you very much for your feedback on our paper. In order to improve our work, we have taken into account the constructive criticisms of the other reviewers. 

Regarding the point about "showing economic benefits": It is beyond the scope of this paper as we do not have enough data to measure and analyse this point. However, we have stated in the introduction that improving efficiency in maintenance can lead to economic benefits. To make this clear, we have included this statement in the paper and marked it with red colour.

Author Response

Reviewer 2:

This paper presents the challenges related human factors of remote maintenance in digitalised production systems. The scope of this paper is actually. However this work is a little confusing, because we can not read the findings from different authors in this area and also we can not find a research gap. 

Response Authors:

Thank you very much for your feedback on our paper. In order to improve our work, we have taken into account the constructive criticisms. For this reason, we have fundamentally improved the content of the article to eliminate ambiguities. All changes are marked in red for better article identification

Reviewer 2: Comments and suggestions are the following:

  • I think that it should be more explained why the authors formulated their results of research experiments only on the basis of the results from 10 companies? Is the studied group a representative group? How was sampling verified? The results on such a small research sample (10 enterprises) should be solidly supported by the explanation that it is a representative group.
  • Moreover, the author stated, that “the experts interviewed comefrom different industries (…), have different roles (…) and represent different technological competences. So, how the the findings can be exploited by future similar works?

Response Authors:

  • Thank you for comments on the sample size and participant expertise for our exploratory interviews. We have reformulated the basic information on the interviewees and presented our professional focus for the experts in more concrete terms.
    Through the insight we gained from experts in the category of manufacturers and users, we were able to obtain realistic details on framework conditions, technological aspects and the application of remote technologies in practice. 
    The adapted passages can be found in the Materials and Methodology section.

Reviewer 2:

  • Please define more precisely and detailed the additional contribution of the research to the recent state of the research field. The discussion must include the results obtained in comparative analysis.
  • The conclusion should be more informative. Again - how the findings can be exploited by future similar works

Response Authors:

  • In order to your Feedback, we changed the results section of the article to focus on the design of a reference model for the evaluation of remote maintenance processes. This has enabled us to emphasise the contribution to research more clearly. The adapted passages are marked in red in the Results and Conclusion sections.

    Thank you again for your constructive feedback. We would be happy to make further enhancements in a second improvement loop. 

Reviewer 3 Report

The essential assessment of a candidate publication is whether the content represents new knowledge that is of general interest. In addition, information regarding a "population", which in this article is industrial companies related to maintenance, requires careful research design, for example, is the selected sample representative? The article contains not sufficient information regarding the informants and the number of informants is far too low. The candidate companies ("the population") for assessment are huge. Remote support is commonly used today and current practice should be consulted first. In addition, experience from similar areas like telemedicine should be consulted, this was an active research area in the 90s.

The technologies discussed in the article seem obsolete, this indicates an outdated understanding of technologies. A novel methodology requires novel technology. For example, reference [11] supports the statement regarding lacking intelligence in technologies. The reference is from 2014. The publication is in German and not available for assessment for most readers of a publication in English. One must assume that the technology is years older. According to Wikipedia Google released its glasses in 2013. My point is that in this area a time span of approximately 10 years is much.

In the text, there are several contrasting statements that are not sufficiently addressed. The importance of maintenance is claimed to be both essential and not essential. That there was no clear understanding among the informant regarding terminology is stated on page 14. This is essential information for the reader and should be stated upfront. Furthermore, it cast a shadow of doubt on the selection criteria and the interview situation. Are the selected candidates representative?

The conclusion states that more research is needed. There is no evidence of this in the preceding text. However, missing standards and interoperation are identified as the main obstacles.

Author Response

Feedback Reviewer 3:

The essential assessment of a candidate publication is whether the content represents new knowledge that is of general interest. In addition, information regarding a "population", which in this article is industrial companies related to maintenance, requires careful research design, for example, is the selected sample representative? The article contains not sufficient information regarding the informants and the number of informants is far too low. The candidate companies ("the population") for assessment are huge. Remote support is commonly used today and current practice should be consulted first. In addition, experience from similar areas like telemedicine should be consulted, this was an active research area in the 90s.

Authors Response: 

  • Thank you for your detailed and constructive feedback. Based on this, we have decided to partially revise the section on methodology. And to explain the small number of interview participants you mentioned. The amended passages are marked in red in the Materials and Methodology section.
    Furthermore, we have also taken your advice regarding the references seriously and increased the Related Work section. 

Reviewer 3:

The technologies discussed in the article seem obsolete, this indicates an outdated understanding of technologies. A novel methodology requires novel technology. For example, reference [11] supports the statement regarding lacking intelligence in technologies. The reference is from 2014. The publication is in German and not available for assessment for most readers of a publication in English. One must assume that the technology is years older. According to Wikipedia Google released its glasses in 2013. My point is that in this area a time span of approximately 10 years is much.

Authors Response:

  • Thank you for your constructed Feedback, we have decided to remove this fact from the paper and add further references to the section related work.
  • The interviews with the experts showed that despite the technologies that have existed for a long time, the companies have not yet been able to establish use cases for practice use. We now go into this more concretely in the Results section. You will find the changed passages marked in red in the article.

Reviewer 3:

In the text, there are several contrasting statements that are not sufficiently addressed. The importance of maintenance is claimed to be both essential and not essential. That there was no clear understanding among the informant regarding terminology is stated on page 14. This is essential information for the reader and should be stated upfront. Furthermore, it cast a shadow of doubt on the selection criteria and the interview situation. Are the selected candidates representative?

Response Authors: 

  • Thank you for your feedback, we have tried to clarify the ambiguities in our current elaboration. We have also included this aspect in the main body of the answer to research question 1. 

Reviewer 3:

The conclusion states that more research is needed. There is no evidence of this in the preceding text. However, missing standards and interoperation are identified as the main obstacles.

Response Authors: 

  • We have tried to implement this aspect due to changes in the focus of the article and concretisation of wording throughout the article. The changes can be seen in the sections marked in red. 

Thank you again for your constructive feedback. We would be happy to make further enhancements in a second improvement loop. 

Reviewer 4 Report

The paper presents an analysis related the remote technologies as common practice in industrial maintenance.

The structure of the paper should also be included at the end of the introductory section.

I think that collecting data from only 10 experts is too little.

How relevant are selected experts across the industry?

Perhaps a statistical analysis of the data collected should be included.

Author Response

The paper presents an analysis related the remote technologies as common practice in industrial maintenance.

The structure of the paper should also be included at the end of the introductory section.

I think that collecting data from only 10 experts is too little.

How relevant are selected experts across the industry?

Perhaps a statistical analysis of the data collected should be included.

Authors Response:

Thank you for your constructive feedback. We have adjusted the structure of the introduction and specifically addressed the components of the article. Changes You will see marked in red.

We have also been able to provide concrete explanations on the number of participants, explaining why the number still leads to well-founded results for our purposes. You will find the concrete explanations in the Methodology section marked in red. 

We take the suggestion to include a statistical analysis but have decided against it due to the time available to us within the framework of our research project.

Thank you again for your constructive feedback. We would be happy to make further enhancements in a second improvement loop. 

Round 2

Reviewer 1 Report

The authors responded to reviewers' comments

Author Response

Thank you for your feedback on our article.

Best regards from the authors.

Reviewer 2 Report

Thank you for your corrections according to my comments.  

But still there is no broader explanation to the questions  “Is the studied group a representative group? How was sampling verified?".

Author Response

Thank you for your feedback on our revised version of the article. 
Regarding your question about the representativeness of the sample and how we checked this, the framework of 10 interviews for a qualitative, exporative study should be seen in the light of Döring & Bortz, 2016. There it says in Chapter 7 Research Design on p. 187: "Qualitative research approach - In the qualitative research approach, open research questions are investigated in great detail on a few research units using unstructured or partially structured data collection methods. The aim is to describe the subject matter and develop a theory. The collected qualitative (non-numerical, i.e. verbal, visual) data are analysed interpretatively".

The research units (the 10 cases/interviews) are described in great detail. This allows a theory to be built using a catalogue of criteria to evaluate successful remote strategies.

We have now formulated this again under the section on data collection. 

Döring, N. & Bortz, J. (with the collaboration of S. Pöschl) (2016). Forschungsmethoden und Evaluation in den Sozial- und Humanwissenschaften [Research Methods and Evaluation in the Social and Human Sciences] (5th completely revised, updated and expanded edition). Heidelberg: Springer. 

Best regards from the authors

Round 3

Reviewer 2 Report

Unfortunately I am not convinced by the research group's explanation and I do not think that these s findings can be exploited by future similar works.

Author Response

Thank you very much for your feedback.

Unfortunately, we did not specifically take into account the aspect that was already mentioned in the first round. We apologise for the omission. We have now added a new paragraph in the Conclusion (highlighted in red) to clarify the wider potential benefits of our findings and hope that this will now be made clear within the paper.

Best regards,
the authors